# New Insights into the Bio-Chemical Changes in Wheat Induced by Cd and Drought: What Can We Learn on Cd Stress Using Neutron Imaging?

**DOI:** 10.3390/plants13040554

**Published:** 2024-02-18

**Authors:** Yuzhou Lan, Genoveva Burca, Jean Wan Hong Yong, Eva Johansson, Ramune Kuktaite

**Affiliations:** 1Department of Plant Breeding, The Swedish University of Agricultural Sciences, P.O. Box 190, SE-23422 Lomma, Sweden; yuzhou.lan@slu.se (Y.L.); eva.johansson@slu.se (E.J.); 2Diamond Light Source Ltd., Harwell Science and Innovation Campus, Didcot OX11 0DE, UK; genoveva.burca@stfc.ac.uk; 3ISIS Pulsed Neutron and Muon Source, Harwell Science and Innovation Campus, Didcot OX11 0QX, UK; 4Faculty of Science and Engineering, The University of Manchester, Alan Turing Building, Oxford Road, Manchester M13 9PL, UK; 5Department of Biosystems and Technology, The Swedish University of Agricultural Sciences, P.O. Box 190, SE-23422 Lomma, Sweden; jean.yong@slu.se

**Keywords:** cadmium, drought, gas exchange, neutron computed tomography, wheat, root architecture

## Abstract

Cadmium (Cd) and drought stresses are becoming dominant in a changing climate. This study explored the impact of Cd and Cd + drought stress on durum wheat grown in soil and sand at two Cd levels. The physiological parameters were studied using classical methods, while the root architecture was explored using non-invasive neutron computed tomography (NCT) for the first time. Under Cd + drought, all the gas exchange parameters were significantly affected, especially at 120 mg/kg Cd + drought. Elevated Cd was found in the sand-grown roots. We innovatively show the Cd stress impact on the wheat root volume and architecture, and the water distribution in the “root-growing media” was successfully visualized using NCT. Diverse and varying root architectures were observed for soil and sand under the Cd stress compared to the non-stress conditions, as revealed using NCT. The intrinsic structure of the growing medium was responsible for a variation in the water distribution pattern. This study demonstrated a pilot approach to use NCT for quantitative and in situ mapping of Cd stress on wheat roots and visualized the water dynamics in the rhizosphere. The physiological and NCT data provide valuable information to relate further to genetic information for the identification of Cd-resilient wheat varieties in the changing climate.

## 1. Introduction

Heavy metal pollution of agricultural soils is a major environmental problem threatening food security across the world. Among the heavy metals, Cadmium (Cd) is extremely harmful as it is known as a threat to living organisms due to its high toxicity and solubility in water [1]. Improper industrial and agricultural activities, e.g., emissions from incinerators, dispersal of mining wastes, the use of contaminated sewage sludges, and the abuse of fertilizers, are regarded as the main sources of Cd contamination in the soil [2]. The Swedish Cd consumption in the world during the 1980s reached 0.8% and Cd was found to originate from NiCd batteries, products with Cd alloys, and artificial fertilizers, where 75% of the average of Sweden’s Cd intake came from agricultural practices [3]. Excessive Cd in soil causes various physiological disorders leading to reduced plant growth: foliar chlorosis [4], impaired photosynthesis [5,6], decreased mineral uptake [7], and transpiration [8]. Additionally, significant negative effects have been reported on tissue phytohormones and root morphological traits due to Cd stress [9].

Being one of the major food crops, wheat is serving as a staple food for more than 50% of the world population [10]. Wheat, compared to other cereals, is known to accumulate Cd in its grains from Cd-contaminated soils through uptake and translocation from the roots [11,12,13]. Therefore, the Cd content allowed in wheat grains to be used for human consumption is regulated in many countries [14,15,16], and in the EU an acceptable limit is 0.1 mg/kg of grain, as set by the commission regulation (EU) 2021/1323. Nevertheless, traces of Cd in food are a threat towards human health and food security, especially when present in wheat-derived products due to the high level of consumption [17,18]. Soil in Skåne, the southernmost province of Sweden, is known to have higher Cd content than soils from other regions, resulting in higher grain Cd content in crops grown on those soils [19]. The intake of Cd from wheat products has been shown to account for a high proportion (~43%) of the total intake of Cd from food in Sweden [20]. Furthermore, soil contamination with Cd has also been documented in a number of countries worldwide, raising great concerns about food security [21,22]. Within the genus *Triticum*, Cd accumulation varies largely among wheat species. In particular, durum wheat has been reported to accumulate a higher amount of Cd than bread wheat, resulting in a higher content in the grain [12,13,23,24]. Therefore, a deeper understanding of the mechanisms behind Cd accumulation in wheat that can potentially improve the risk management of Cd in wheat production and the breeding of Cd-resilient wheat is extremely urgent.

Wheat productivity is increasingly impacted by drought stress, not least due to climate change contributing to a global increase in temperature and spells of extreme weather events [25,26,27]. Drought induces several changes in the plant, including physiological, biochemical, morphological, and molecular changes [28,29,30]. Yield losses from drought events have been reported for wheat across many regions around the world [25,26,27]. During the drought period, the roots of the plant are an organ that directly encounters drought stress in the soil, and, therefore, wide attention has been given to understanding the response of root traits to drought. In general, severe adverse effects from drought stress have been reported on root growth in wheat [31].

As global climate change will continue to contribute to an increasing amount of abiotic stress events, plants will be subjected to various combinations of abiotic stresses, e.g., drought, heat, and heavy metals, as well as soil compaction [32]. Few studies have evaluated the effects of combined stress, e.g., drought and Cd, on wheat development and production. However, drought stress has been reported to reduce the Cd uptake from soil in *Ricinus* and *Brassica* species [33]. The drought-restricted root growth and capacity to absorb nutrients and trace metals might be the causes of the reduction in Cd uptake in severe growing environments [34]. Changes in root characteristics under drought have been proven to decrease Cd accumulation in castor beans [35], while very limited information is available on drought impact and Cd accumulation in wheat.

As both Cd and drought are environmental abiotic stresses based on changes in soil properties, understanding root–soil interactions is crucial for combined studies on the effects of drought stress and long-term strategies (including breeding) to reduce Cd uptake in Swedish wheat and wheat worldwide. The conventional method of measuring root traits is based on a physical separation of the root samples from soil [36], which is disruptive to the rhizosphere and imprecise, as it leads to a loss in root volume, and this approach is also highly labor-intensive and time-consuming. Since the late 1970s, a non-invasive soil imaging methodology has arisen with the development of novel technologies, including X-ray computed tomography (XCT) [37,38], magnetic resonance imaging (MRI) [39], nuclear magnetic resonance imaging (NMR) [40,41], and neutron imaging (NI) [42,43,44]. The advent of neutron computed tomography (NCT) has facilitated novel root–soil research as it provides an in situ visualization of the rhizosphere in a 3D form enabling greater biological insights than a traditional flatbed 2D observation. The water distribution and movements between the roots and soil have been successfully explored using NCT [45,46]. Furthermore, the root length of wheat quantified using NCT has been proven to be significantly correlated with root length measured using physical flatbed scanning [47]. Most previous studies focusing on Cd stress and wheat root traits have been based on either flatbed scanning or biomass weighting [48,49,50]. To our knowledge, there is no information on visualizing the wheat root morphology during Cd stress using NCT.

Thus, the aim of this study was to evaluate the individual Cd and combined effects of Cd and drought on the early growth of durum wheat in two different growth media, i.e., soil from a field in southern Sweden and commercial sand. We also questioned whether NCT can be useful to obtain information about the Cd stress impact on wheat development and water distribution in soil vs. sand. For the first time, this study shed light on how the physiological methods and non-invasive NCT can be combined to provide valuable information on durum wheat development under stress conditions and for the first time show root–soil/sand interactions in wheat grown under Cd stress.

## 2. Results

### 2.1. The Effects of Cd and Cd + Drought on Gas Exchange Parameters

A significant effect was found on gas exchange parameters, such as the *A/Ci* curves, stomatal conductance, and water use efficiency (WUE) from the control in the soil (CSoil), Cd concentration in the soil (CdSoil) and Cd concentration + drought (CdDSoil; Figure 1). A content of 60 mg/kg Cd in the soil combined with drought (CdDSoil) reduced the *A/Ci* curve intensity significantly as compared to CSoil conditions (Figure 1a), while an additional and significant decrease in A values was observed when the Cd concentration increased (from 60 to 120 mg/kg) in both the no drought and drought conditions (Figure 1a,b). Thus, the largest effect on the A/Ci curve was obtained for the highest Cd concentration (120 mg/kg) combined with drought (CdDSoil), clearly indicating the inhibitory cross effect of Cd and drought on photosynthesis in wheat plants (Figure 1a,b).

After a growing period of 21 days (four days in 120 mg/kg of Cd + drought conditions), the plants showed clear stress symptoms, e.g., CdDSoil plants were smaller in size than CSoil plants, while no such clear differences were observed between CdSoil and CSoil plants (Figure 1c). Cadmium in the soil (both at 60 and 120 mg/kg) had a significant effect on the stomatal conductance, while neither the difference in Cd concentration in the soil nor the addition of drought conditions to the Cd-treated plants showed a significant influence on the stomatal conductance of the plants (Figure 1d). The transpiration rate (*E*) and water use efficiency (WUE) showed a similar pattern, where the combination of Cd in the soil (both 60 and 120 mg/kg) and drought conditions (i.e., CdDSoil) resulted in a significant decrease in both *E* and WUE (Figure 1e,f). No significant effects were obtained for CdSoil-treated plants as compared to CSoil plants at the studied Cd amounts.

### 2.2. The Levels of Cd in Wheat Tissues under Cd and Cd + Drought Stresses

The Cd content in dried shoots and roots, as evaluated using inductively coupled plasma–mass spectrometry (ICP-MS), showed differences for plants grown in sand versus soil (Figure 2). Plants grown in sand accumulated significantly more Cd in both their roots and shoots than plants grown in soil, independent of whether the plants were also drought-treated. We observed Cd contents 15 and 6 times higher in CdSand roots (1140 mg/kg) and shoots (170.5 mg/kg) than in CdSoil roots and shoots (Figure 2a), while CdDSand roots (582 mg/kg) and shoots (60 mg/kg) were approximately 10 and 2 times higher than those found in CdDSoil, respectively (Figure 2a).

A significantly lower shoot dry weight was obtained for CdDSoil and CdDSand samples as compared to plants grown in CSoil and CSand (Figure 2b). No significant effect on the dry weight of roots was obtained for plants grown in soil, while the plants grown in sand showed a significantly lower root dry weight under the sole treatment of Cd (Figure 2b).

### 2.3. Foliar Chlorophyll Content and Fluorescence under Cd and Cd + Drought Stresses

A significant reduction in the foliar chlorophyll content (SPAD) was obtained in CdDSoil and CdDSand samples as compared to control samples (CSoil and CSand; Figure 3a). No significant difference in the SPAD value was observed in the plants grown under sole Cd stress compared to control samples (CSoil and CSand; Figure 3a). No significant difference in the fluorescence (Fv/Fm) value was found among plants from the different treatments (Figure 3b).

### 2.4. Effects of Growing Media on Root Volume and Architecture as Evaluated Using NCT

The NCT study clearly showed a variation in root volume for the two wheat genotypes, Tramadur and Duramonte, grown in different media (Figure 4). A higher root volume in sand (Duramonte: 57.1 mm^3^, Tramadur: 45.4 mm^3^) than in soil (Duramonte: 32.9 mm^3^, Tramadur: 36.5 mm^3^) was observed for both genotypes (Figure 4a). Duramonte exhibited a larger root volume in sand as compared to soil, different to what was observed in Tramadur. A clear difference in the 3D root architecture between the soil and sand samples was observed using NCT for both genotypes (Figure 4b,c compared to Figure 4d,e). In soil, roots were less branched and relatively straight, and were distributed along the tube from the top down to the bottom (Figure 4b,c). The roots in sand were somewhat more branchy, dense, and more concentrated around the middle of the tube with more bends and sharp turns than in soil (Figure 4d,e).

### 2.5. Root Volume and Architecture of Wheat under Cd Stress Evaluated Using NCT

The root volume and branching of Tramadur plants under Cd stress clearly varied among the sand and soil samples, as shown using NCT (Figure 5a). In both soil and sand, the root volume decreased due to Cd stress from 57.7 mm^3^ (control) to 29.2 mm^3^ (Cd sample) in soil and from 63.7 mm^3^ (control) to 58.0 mm^3^ (Cd sample) in sand (Figure 5a). The 3D branching of roots was somewhat different between the soil and sand samples, and smaller root branching most likely due to Cd stress was observed only in soil samples when compared to non-stress conditions (Figure 5b,c). In sand, the roots were more developed and showed a curly and more branching pattern as compared to soil, although sharp turns of roots remained unchanged between the control and the Cd stress samples (Figure 5d,e).

### 2.6. Water Distribution in Different Growing Media as Monitored Using NCT

It was possible to differentiate the water distribution between the growing media (soil and sand) and a varying pattern of the “roots-growing media-water” as illustrated in 3D NCT scans shown in Figure 6. In soil, water was rather evenly dispersed with discontinuous water droplets along the soil particles from the top to the bottom of the tube (Figure 6a,c,d). In sand, the water was largely localized in the lower part of the column, while in the upper part, large water regions were formed (Figure 6e,g,h; light grey continuous color). A large continuous water network was closely associated with the pores observed within the sand media (Figure 6g,h). In soil, the column displayed the small water areas that were evenly dispersed within the soil aggregates together with channels across the whole tube within a range of 1 mm up to 5 mm (Figure 6d).

As expected, the growing media porosity differed between the soil and sand, mainly in the packing of media particles, as numerous aggregated particles with pores along them were observed in soil compared to sand (Figure 6b compared to Figure 6f). The soil particle aggregates were approximately 1 mm in size after root penetration and water treatment, and formed a robust soil matrix (Figure 6a,d; dark grey and black areas, respectively). The sand media consisted of a rather compact media system in which the particles with reduced aggregates, smaller pores, and greater moisture content were observed in comparison with the soil system (Figure 6e,f).

## 3. Discussion

The present study clearly showed that a high soil Cd concentration (e.g., 120 mg/kg) and, realistically, a combination of soil Cd with drought stress, negatively affected most of the photosynthetic parameters (e.g., stomatal conductance and relative chlorophyll content) and wheat plant development. This is in agreement with previous studies on Cd as a sole stress factor, where at high Cd concentrations decreased plant growth was found due to negative effects on photosynthesis oxidative stress in cells and the water balance [51,52,53]. Although the Cd stress effect differed in various growth media, e.g., soil-based vs. hydroponic growing systems [54], we are also aware that the chosen Cd concentrations in this study can be somewhat elevated compared to the real soil concentration expected in the field.

From the present study, drought was found to add to the effects of soil Cd as a sole factor on foliar stomatal conductance and water use efficiency (WUE), resulting in a reduction in the growth and photosynthesis of wheat. The water use efficiency (WUE) is known to be an important parameter to evaluate the plant physiological response and tolerance to drought [55,56]. The fact that the WUE was severely depressed by the combined Cd and drought stresses (with no individual effect and regardless of Cd concentration) suggests that the drought stress impacted the photosynthesis rate (*A*) more than the transpiration rate (*E*). Since biomass production in wheat is highly linked to transpiration, a focus on breeding is therefore important to identify genotypes with a high capacity to capture moisture in the soil, which is a good opportunity to improve the yield under drought stress conditions [57] in future studies. The limited transpiration rate under drought stress noted for the wheat plants in this study may be the result of hydraulic restrictions within the plant, which has in previous studies been shown to slow down the water moving from the roots to the upper leaves [58]. The foliar photosynthetic rate decrease during drought in the present study has in previous studies been found to be related to several mechanisms such as decreased turgor pressure, stomata closure, a reduction in CO_2_ assimilation, and the lowering of photosystem I (PSI) and II (PSII) activities [59,60], which could be part of the explanation in this study. Thus, the highly hampered photosynthetic characteristics and growth of wheat plants at early development stages from high soil Cd concentration (120 mg/kg) combined with drought in this study was the result of the root-to-shoot translocation mechanism of Cd. This translocation mechanism seems to differ and was strongly impacted by the growth media used (the roots in the sand were prone to accumulating more Cd than those in the soil).

The general Cd levels in agricultural soils range between 0.1–1.0 mg/kg [61], and in this study we used a high Cd concentration to imitate heavily contaminated site soils recorded in different regions, e.g., 32.5 mg/kg in Deyang, China [62], 0.05–176 mg/kg in Gebze, Turkey [63], and 60–100 mg/kg in Vallåkra, Trelleborg and Österlen, Sweden [64]. Based on the fact that the Cd is taken up through roots, for further translocation to other parts of a plant (e.g., shoots and grains) [7,65], the used soil Cd levels were assumed to have a severe phytotoxic effect on wheat tissues through an inhibiting effect on morphological and physiological processes, as previously described [66]. Taking this into account, no significant differences in the final biomass of the wheat plants were observed in the present study, neither for dried roots-shoots grown in soil or sand nor compared to their respective control. This was most likely the result of the short experimental growth period (21 days) in this study. It is noteworthy that the Cd accumulation in roots was more than 10 times higher in sand than in soil regardless of the applied stress treatments (e.g., individual Cd or a combination of Cd + drought stress). These observations are indicative of different root-to-shoot Cd translocation mechanisms for the growth media used, which was supported by other studies. For example, the Cd uptake by plant roots has in previous studies been shown to be greatly determined by the properties of the growing media [67]. In soil, the Cd chemical forms might be water soluble; exchangeable (bioavailable and mobile) with various linkages to carbonate, oxide, and organic compounds; or existing independently as residual compounds [68]. Furthermore, the bioavailability of Cd has been found to be more relevant to plant Cd uptake than the total soil Cd content [69]. Due to the complexity of Cd–soil interactions that are influenced by pH, redox potential (Eh), organic matter (e.g., clay minerals), and soil microorganisms [70,71,72], we assume that the Cd impacts on plants and the Cd–soil interaction might differ during various field conditions. Interestingly, the studied soil and sand media dispersions in water (results not shown) showed no significant difference in pH, suggesting factors such as Eh, organic matter, and soil microorganisms were playing a minor role in this study. Cadmium in the soil-water dispersion are known to exist in the form of free Cd^2+^ in high Eh conditions, while it is precipitated as CdS and CdCO_3_ in low Eh conditions [73,74]. In the present study, the level of Cd in the soil taken from the field was 0.33 mg kg^−1^ of Cd (its chemical form in the soil was unclear). As soil organic matter can function as an adsorbent to complex Cd^2+^ and may result in a reduction of Cd bioavailability [75,76], this might partly explain differences in the root-Cd content of wheat plants grown in sand and soil. To complicate the story about Cd availability in the soil further, the application of chelated fertilizers has the potential to result in the accumulation of both synthetic and natural organic substances such as ethylenediaminetetraacetic acid (EDTA) and vegetable-extracted amino acids in the soil, and these chelates can also play a role in enhancing the bioavailability to plants by forming metal chelates [77,78,79], as was seen in ryegrass [80]. However, adding to this, in our study we can speculate about suitable soil microbes that can decrease Cd bioavailability by binding Cd with their secreted proteins and further converting it into non-available forms. Similar findings were observed in another study [81]. In our study, one explanation can be the presence of organic matter (acting as Cd^2+^ adsorbents), and microbes in the soil that might be responsible for lowering the Cd availability. Interestingly, higher Cd immobilization in soils was observed for wheat and rice when higher levels of organic matter (biochar) were added [78].

When comparing individual Cd and Cd + drought stresses, a significantly lower Cd accumulation was found in plants grown in sand with a combination of Cd + drought, suggesting the clear impact of a water imbalance (low availability) and potential mechanisms related to the mitigation of drought-induced damages in plants. The impact of Cd + drought has shown that drought reduced the Cd content in both shoots [82] and roots [83] in several plant species (e.g., peanuts, *Ricinus communis*, and *Brassica juncea*), and also showed weakened soil-to-root translocation of Cd under drought [33,35]. Nevertheless, for the common bread wheat grown in Cd-contaminated soils in a field, the results were contrary and indicated an increase in Cd accumulation observed under drought [52]. For durum wheat grown in the field containing elevated amounts of soil-Cd in Sweden (Österlen), drought seemed to negatively impact the production of durum wheat and increase Cd accumulation in the grains. From this and other studies, it can be assumed that the impact of drought on Cd accumulation is determined by various factors such as the plant species, growth stage and media, stress level, and duration of stress period [33,35,83].

The 3D root architecture results showed that detailed wheat root growth information can be quantified and visualized in non-transparent media using the non-invasive and promising NCT technique. This observation can justify our hypothesis that NCT can be a valuable technique to study Cd stress in wheat. The root volume, mapped using NCT, of the two wheat genotypes (Duramonte and Tramadur) suggested a similar root vigor with some differences in root volume in soil as compared to sand between the genotypes. From the 3D images of wheat plants growing in the soil, Tramadur seemed to develop more lateral roots at the bottom of the tube, which differed from Duramonte. In sand, the roots of both genotypes seemed to be clearly more contorted than in soil, with more volume accumulated in the middle of the tubes. These morphological changes (e.g., alteration in growth orientation) of root systems were influenced by the soil bulk density and soil type, and it is widely accepted that a more compacted soil texture leads to greater tortuosity in root growth [84,85,86]. However, some studies have also noted reduced root length and concomitant thickening of roots with increasing soil strength [87,88]. No studies have until now evaluated the wheat root growth paths and its differences in various growth media, such as soil and sand, which we revealed for the first time in this study using NCT. Based on our NCT-derived images, we interpreted that the twisted shape of roots observed in this study was due to the stronger resistance force that roots encountered during elongation in sand. More detailed research is needed to evidence whether wheat roots can actively alter their growing orientation while seeking more soil nutrient resources, as indicated in previous studies [89], and avoiding Cd and arid patches in the rhizosphere.

The results from NCT 3D images also clearly showed reductions in the root volume under Cd stress as compared to their respective controls in soil and sand. This is the first quantification of wheat root volume and characterization of the wheat root architecture under Cd stress using the NCT technique. Furthermore, this study indicates the unique potential of neutron imaging as a non-destructive method for quantitative mapping of Cd stress on plants’ roots that can help with Cd management in Swedish soils. A more profound study is needed for the further exploration of NCT towards critical Cd transport in different parts of wheat plants. From the 3D images, wheat roots appeared thinner and with fewer lateral roots under Cd stress in soil, which corresponded to results from previous studies showing a significant reduction in the root diameter, lateral roots, and root volume under Cd stress [90].

Neutron imaging has been used to investigate the soil water distribution previously [44,47,91] and the results from this study showed a similar water pattern in wheat roots in soil. This water pattern differed greatly compared to the water pattern in sand, which clearly indicated an impact on root growth from the growth media. Differences in soil particle sizes were recorded from the 3D images in this study. Compared to the originally sieved particle size of 1 mm, the larger aggregates of soil were formed with root and water penetration, resulting in larger air pores and an increased porosity of the soil column. The faster moisture loss observed in soil might potentially be explained by this aerated soil structure, as well as a different soil vs. sand architecture. In this study, both media were surface irrigated, so the different physical properties between soil and sand might account for the water distribution differences observed. Soil aggregates are known to absorb water, and the absorption capacity and water retention ability are greatly dependent on the soil aggregate sizes [47]. Unlike soil, sand particles do not absorb moisture in the same way, which resulted in the outcome observed in this study, justifying the importance of growing media in facilitating water retention.

## 4. Materials and Methods

### 4.1. Plant Materials

Two durum wheat (*Triticum durum* L.) genotypes, Tramadur and Duramonte, were provided by Lilla Harrie Valskvarn AB, Kävlinge, Sweden, and (i) a greenhouse trial with Tramadur was carried out at the Department of Plant Breeding, the Swedish University of Agricultural Sciences (SLU), Lomma, Sweden, while (ii) a neutron imaging experiment as a proof of concept with both Tramadur and Duramonte was performed at the IMAT beamline (Imaging and Materials Science & Engineering) [92] from the ISIS Pulsed Neutron and Muon Source, UK. In both cases, wheat seeds were sown 10 mm beneath the media surface in quartz tubes in order to establish the plant-growth media settlement suiting for neutron imaging experiment (e.g., the choice of tube type and size was adapted to the beamline technical requirements) to monitor Cd stress on wheat development. Three seeds of each variety were planted in individual tubes and these were used as replicates. The proof of concept experiment was performed on the highly selected plant material that was subjected only to Cd stress due to the hypothesis testing whether NCT can provide valuable information on the wheat plant development (drought stress experiment was excluded due to long data collection times and severe plant physiological changes).

### 4.2. Soil and Sand

Two types of cultivation media, i.e., clay loam soil from a field of Southern Sweden (55°43′3.85″ E and 13°7′51.19″ S), provided by Lilla Harrie Valskvarn AB, Kävlinge, Sweden, and commercial sand purchased from Rådasand (Lidköping, Sweden), were used to grow wheat plants in both cases. The field soil sample was HNO_3_-digested (120 °C, 30 min) to measure Cd concentration originally contained 0.33 mg kg^−1^ of Cd as analyzed using Inductively Coupled Plasma Atomic Emission Spectrometry (ICP-OES) [93]. The soil was passed through a 1 mm sieve before sowing the seeds, and the particle size of the sand was 0.8–1.2 mm. The soil characteristics from Österlen region were of a varying type that included to a large extent moraine and hummus of acidic nature [94].

### 4.3. Greenhouse Experiment for Physiological Traits

#### 4.3.1. Growing Conditions and Treatments

The greenhouse trial was conducted in February in a greenhouse chamber (Alnarp, Sweden) at 25 °C using artificial light to achieve a consistent 12 h daylength (7:00–19:00). Plants were grown in common glass tubes (SCHOTT, Mainz, Germany) with either soil or sand filled to 80 mm height in order to mimic growing conditions and create wheat plant-growth media suiting system for neutron imaging experiment. Wheat plants were subjected to non-stress control growing conditions in soil and sand (CSoil and CSand), Cd stress (CdSoil and CdSand), and Cd + drought stress (CdDSoil and CdDSand) growing conditions; single drought conditions were not included in the study due to main focus on Cd.

The water content in CSoil, CdSoil, CSand, and CdSand was maintained using surface watering with 3 mL water every second day. The drought stress was applied by withholding the water 10 days after sowing (DAS) until the end of the experiment in CdDSoil and CdDSand. The Cd solution used contained 2.55 mg/mL of cadmium nitrate, Cd (NO_3_)_2_. In soil, the cadmium levels (60 and 120 mg/kg) were used to grow CdSoil and CdDSoil samples, and these levels were obtained with Cd(NO_3_)_2_ solution (adjusted to Cd natural level in the soil). The Cd (NO_3_)_2_ solution was applied on 14 DAS and 18 DAS, separately. In sand, the cadmium levels of 30 mg/kg and 60 mg/kg were used for sand samples CdSand and CdDSand, and Cd(NO_3_)_2_ solution was, similarly as for soil cultivation, applied on 14 DAS and 18 DAS, separately. Three biological replicates of wheat samples were used.

#### 4.3.2. Gas Exchange Measurements

Gas exchange parameters, i.e., net photosynthesis rate (*A*), intercellular CO_2_ concentrations (*Ci*), transpiration rate (*E*), and stomatal conductance (*g*) of wheat plants grown in soil were measured on young and fully expanded leaves after 17 and 21 DAS in three replicates with two plants in each replicate using a portable photosynthesis system (LI-6800, Li-Cor Inc., Lincoln, NE, USA) [95]. Due to the small leaf size, each gas exchange measurement was performed using two wheat leaves (combined leaf area measured earlier; Figure 7). To improve the precision, the gas exchange instrument was calibrated one day before each measurement against the ambient temperature and humidity, with regular chamber-matching routine every hour. The *A*/*Ci* curves were plotted based on *A* and *Ci* using the reference CO_2_ concentration at 400, 200, 100, 50, 0, 200, 400, 600, 800, 1000, 1200, and 1400 µmol mol^−1^. Data of g were collected under fixed reference CO_2_ concentration of 400 µmol mol^−1^ and fixed light intensity (*Q*) of 1000 µmol m⁻^2^ s⁻^1^. The instantaneous water use efficiency (WUE) was calculated as the ratio of *A* to *E*, where *E* is the transpiration rate (mol_water_ m^−2^ s^−2^) [96]. To ensure the validity of gas exchange experiment and data collection, all the measurements were performed from 09:30 to 16:30, when leaf stomata remained naturally open. Due to the short measuring window, wheat plants grown in sand were not subjected to gas exchange measurements.

#### 4.3.3. Foliar Chlorophyll Content and Chlorophyll Fluorescence

The foliar chlorophyll content (MC-100 chlorophyll concentration meter, Apogee Instruments, Logan, Utah, USA) represented by soil plant analysis development (SPAD) value and chlorophyll fluorescence (Pocket PEA chlorophyll fluorimeter, Hansatech Instruments, King’s Lynn, UK) represented by ratio of variable fluorescence to maximum fluorescence (Fv/Fm) value were measured on the same young and fully expanded leaves during 09:30 to 16:30 in both soil and sand 21 DAS (three days after the second Cd treatment) in three biological replicates, similarly as reported earlier [97,98].

#### 4.3.4. Dry Weight and Cd Content

At 22 DAS, all plant samples were collected and each seedling was sampled separately in two parts, i.e., shoots and roots. Both shoots and roots were dried in an oven (Memmert, Germany) at 70 °C for 24 h, and later three biological replicates of the dry biomass were weighed. The two biological replicates of each sample were digested with HNO_3_ at 120 °C for 30 min and the Cd content was measured using inductively coupled plasma–mass spectrometry (ICP-MS).

### 4.4. Root Architecture Assay Based on NCT

#### 4.4.1. Growing Conditions and Treatments

Wheat plants were grown in bottom-capped boron (B) free quartz tubes (Donghai Baosheng Quartz Products Co. Ltd., Donghai, China) with an inside diameter of 17 mm, an outside diameter of 20 mm, and a length of 100 mm in a simplified growth chamber at consistent temperature and daylength (22 °C and 12 h, respectively; growing settings were chosen according to the technical set-up of the beamline). Selected growth medium (soil or sand) was filled in tubes and up to the 80 mm mark. Wheat plants were grown under non-stress “control” conditions in either soil or sand (CSoil and CSand), and in medium containing Cd (CdSoil and CdSand), and two replicates were used. At 15 DAS, external Cd treatment (120 mg/kg) was applied by adding Cd(NO_3_)_2_ solution to CdSoil and CdSand tubes. Selected plants from each treatment were subjected to the NCT measurement.

#### 4.4.2. NCT for Root Analysis

The NCT measurements were carried out at the IMAT neutron imaging beamline of the ISIS neutron spallation source at the Rutherford Appleton Laboratory (UK) [99] using an optical system equipped with a CCD Andor camera for the standard neutron tomography based on a previously described protocol [47,99,100]. The scans were performed on 14 and 16 DAS of wheat plants. At 14 DAS, NCT was performed on two wheat plants grown in soil and sand of both Tramadur and Duramonte genotypes (before the first Cd treatment) in order to evaluate the method’s suitability and testing the hypothesis whether the method can provide suitable information on root morphology variation induced using Cd stress. For further NCT analysis, Tramadur was selected and used in Cd treatment (due to technical feasibility of the experiment). At 16 DAS, NCT was performed on the Tramadur genotype in order to investigate the effects of Cd treatments in two growing media (CSoil, CdSoil, CSand, CdSand).

#### 4.4.3. Image Reconstruction and Root Segmentation

Raw images were reconstructed into a stack of TIFF format files using the software Octopus Reconstruction, version 8.9 [101,102]. The reconstructed images were then imported into the software 3D slicer for 3D architecture generation, segmentation, and analysis. The root segmentation was performed manually due to the low contrast in neutron attenuation values between some sections of the root system and the moisture of soil/sand. The relative volumetric moisture content of the soil/sand was then compared based on the relative neutron attenuation [46].

### 4.5. Data Analysis

Means from the gas exchange experiments which include chlorophyll content and chlorophyll fluorescence, as well as means of dry weight and Cd concentration were compared using Tukey’s post hoc test in the software R Studio, version 2023.06.1 [103] with the R package “agricolae”. Averaged NCT data from each scan were also used.

## 5. Conclusions

As a consequence of more frequent climate change-linked perturbations, occurrences of Cd and drought scenarios are challenging food supplies by lowering wheat production. The Cd + drought stress scenario used in this study is likely to prevail in the future, thereby severely impairing the net photosynthesis rate of wheat. We suggest that the higher Cd levels in sand-grown roots than in soil were due to different medium-root-to-shoot transfer mechanisms. For the first time, unique 3D information about the roots’ architecture during Cd stress was obtained, which highlighted the good potential of NCT as a non-destructive method for both the quantitative mapping of Cd’s effect on wheat roots and the water dynamics in sand and soil. The medium-related rhizospheric differences determined using NCT were related to the Cd levels, particle size variation, and porosity of the growing media, which were likely in turn to influence the wheat root architecture. In addition to root morphology, NCT images successfully visualized novel three-dimensional structural differences in the water distribution between soil roots and sand roots, which might help to better understand and model the root–substratum–water interactions. Further work is needed to identify suitable wheat genotypes that could be cultivated with selected soil microbes and organic amendments in Cd-contaminated Swedish soils, and produce wheat grains that are fit for human consumption, i.e., with an acceptable Cd content.

## Figures and Tables

**Figure 1 plants-13-00554-f001:**
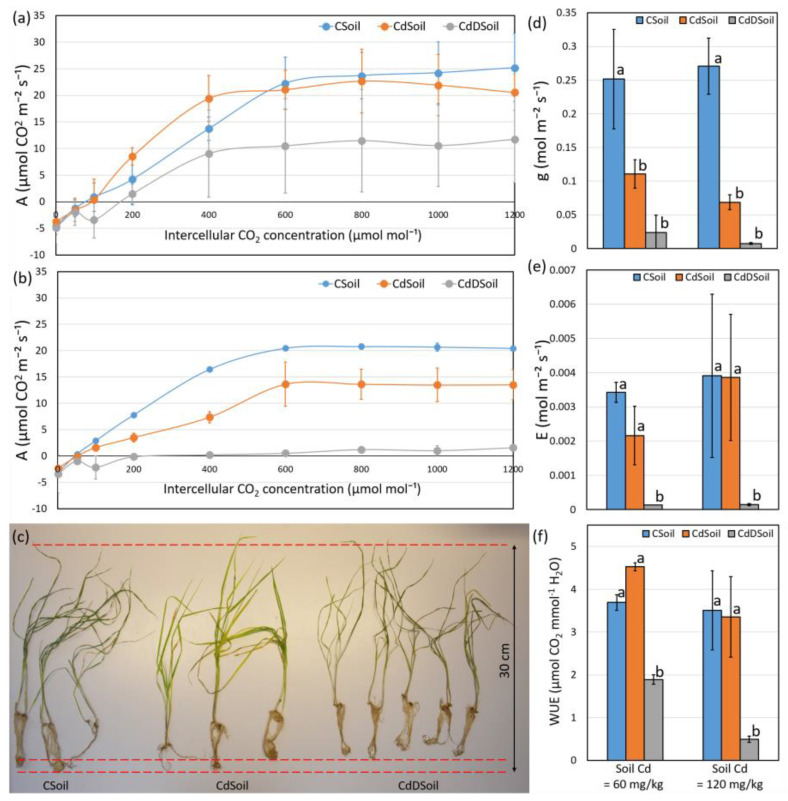
Effects of Cd and Cd + drought on gas exchange parameters and plant development. (**a**) *A*/*Ci* curves of net photosynthesis (*A*) vs. intercellular CO_2_ concentration (*Ci*) of plants grown at control (CSoil) and 60 mg/kg of Cd for both no drought (CdSoil) and during drought (CdDSoil); (**b**) *A*/*Ci* curves of plants grown in CSoil and at 120 mg/kg of Cd for both no drought (CdSoil) and during drought (CdDSoil); (**c**) wheat genotype Tramadur after 21 days of growth in CSoil, CdSoil (four days under 120 mg/kg of Cd), and CdDSoil (four days under 120 mg/kg of Cd); (**d**) stomatal conductance (*g*) at 400 µmol mol^−1^ of CO_2_ concentration and 1000 µmol m^−2^ s^−1^ of light intensity (*Q*) in CSoil, CdSoil, and CdDSoil under 60 mg/kg and 120 mg/kg of Cd; (**e**) transpiration rate (*E*) at 400 µmol mol^−1^ of CO_2_ concentration and 1000 µmol m^−2^ s^−1^ of *Q* in CSoil, CdSoil, and CdDSoil under two levels of Cd treatments; (**f**) instantaneous water use efficiency (WUE) at 400 µmol mol^−1^ of CO_2_ concentration and 1000 µmol m^−2^ s^−1^ of *Q* in CSoil, CdSoil, and CdDSoil under two levels of Cd treatments. All the data are presented in the form of mean values of three biological replicates accompanied with an error bar representing standard deviation. Different letters indicate significant difference detected using Tukey’s post hoc test at the level of *p* < 0.05 between the CSoil, CdSoil, and CdDSoil samples under the same Cd treatment level.

**Figure 2 plants-13-00554-f002:**
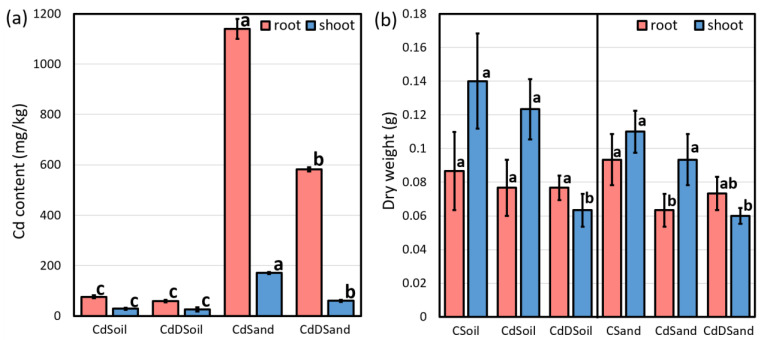
Cd content in 21-day-old (four days under 120 mg/kg of Cd) durum wheat root and shoot samples (Tramadur) grown under Cd and Cd + drought stress in soil (CdSoil and CdDSoil) and sand (CdSand and CdDSand) (**a**), and dry weight of samples grown under control (CSoil and CSand), Cd stress (CdSoil and CdSand), and Cd + drought (CdDSoil and CdDSand) conditions (**b**). Different letters indicate significant difference detected using Tukey’s post hoc test at the level of *p* < 0.05 between CSoil, CdSoil, and CdDSoil samples in either root or shoot.

**Figure 3 plants-13-00554-f003:**
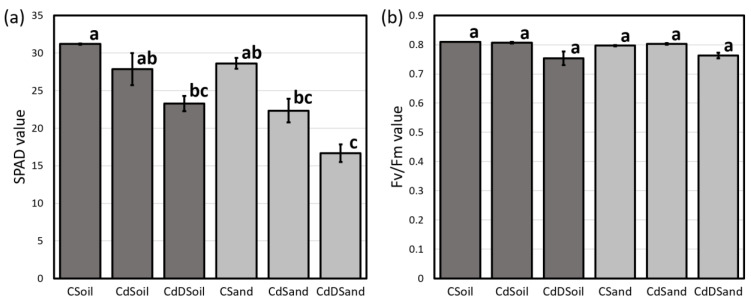
The effects of cadmium (120 mg/kg) and drought treatments on wheat foliar chlorophyll characteristics. The foliar relative chlorophyll content measured as the soil plant analysis development (SPAD) value (**a**) and ratio of variable fluorescence to maximum fluorescence (Fv/Fm) value (**b**) of wheat genotype Tramadur grown in two media. The treatments were as follows: control (CSoil and CSand); Cd stress (CdSoil and CdSand); Cd + drought stress (CdDSoil and CdDSand). Different letters indicate significant difference detected using Tukey’s post hoc test at the level of *p* < 0.05 between treatments.

**Figure 4 plants-13-00554-f004:**
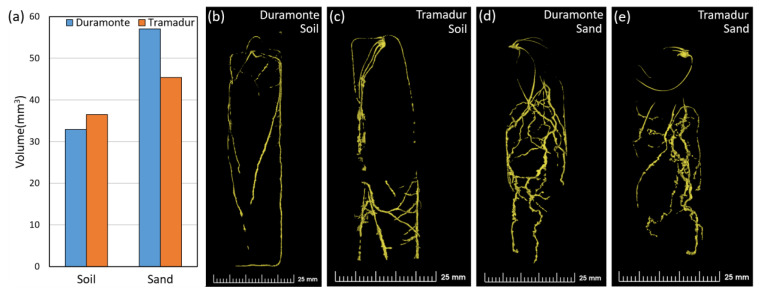
The root volumes of two wheat genotypes, Duramonte and Tramadur, as evaluated using neutron computed tomography at 14 days after sowing in different growing media (**a**) and the 3D structure of genotype Duramonte (**b**,**d**) and genotype Tramadur (**c**,**e**) grown in either soil or sand.

**Figure 5 plants-13-00554-f005:**
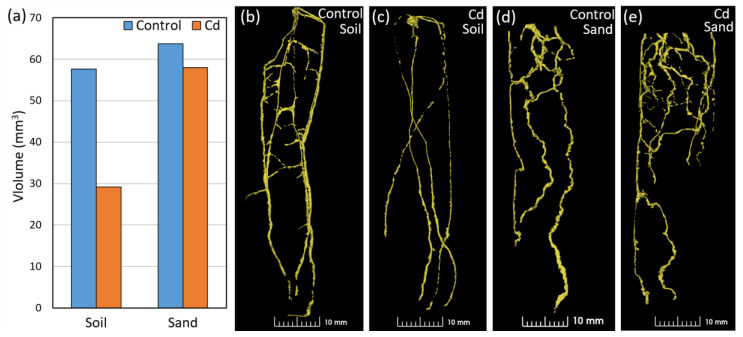
The root volumes of wheat plants (genotype Tramadur) evaluated using neutron computed tomography at 16 days after sowing in different growing media (**a**) and the 3D structure highlighting the architecture of root samples in control (**b**,**d**) and under 120 mg/kg of Cd stress (**c**,**e**) grown in either soil or sand.

**Figure 6 plants-13-00554-f006:**
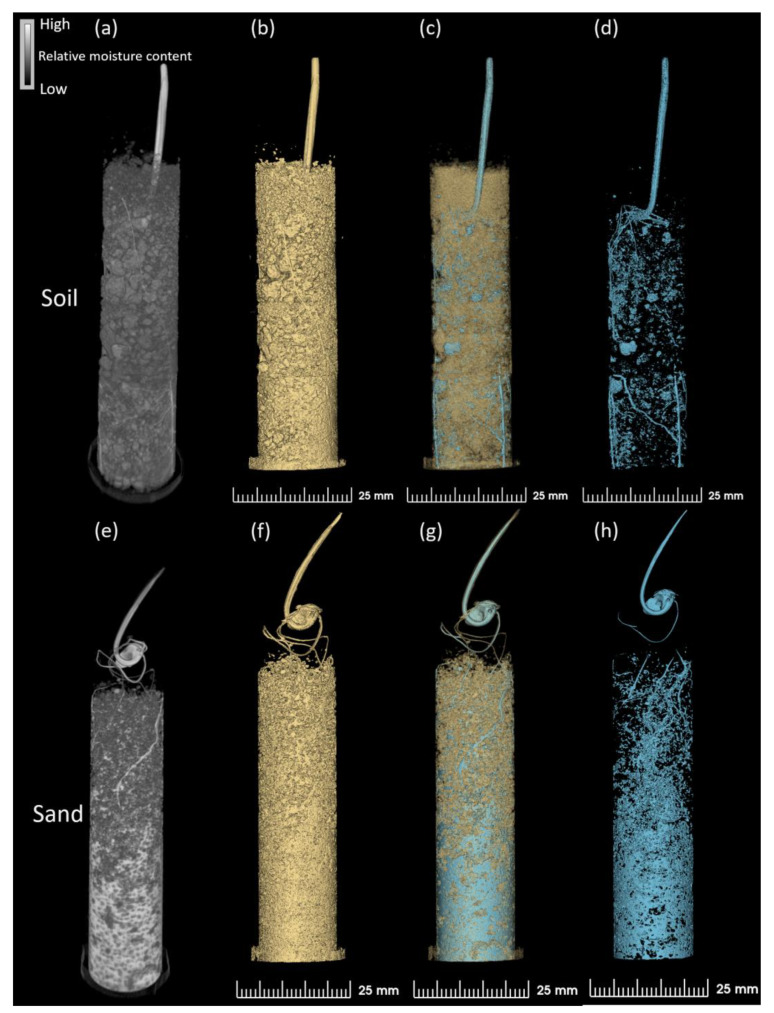
Neutron computed tomography images of wheat plants (genotype Tramadur) showing the media and relative moisture distribution projections in soil (**a**) and sand (**e**). The reconstructed 3D images of soil (**b**) and sand (**f**), media + water (**c**,**g**), and water (**d**,**h**) in either soil or sand growing conditions.

**Figure 7 plants-13-00554-f007:**
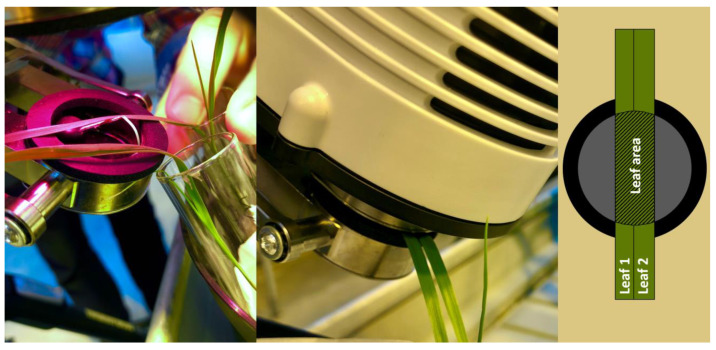
Illustrations for the two-leaf system used in the gas exchange measurement and the combined leaf area used for calibration.

## Data Availability

The data are openly available from the first author and can be accessed upon reasonable request.

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
