# Peer review of "New Insights into the Bio-Chemical Changes in Wheat Induced by Cd and Drought: What Can We Learn on Cd Stress Using Neutron Imaging?"

_plants, 2024, doi:10.3390/plants13040554_

Round 1

Reviewer 1 Report

Comments and Suggestions for Authors

The present manuscript explored the impact of Cd and Cd+drought stress on durum wheat growing in two types of media, soil and sand, probing two Cd levels. The authors studied gas exchange parameters, levels of Cd in wheat tissues, and leaf chlorophyll content and fluorescence. Moreover, they explored the root architecture using a non-invasive neutron computed tomography (NCT).

The results are interesting, showing the effects of Cd and Cd+drought on the parameters tested and presenting an innovative method by NTC to analyze the effects of Cd on wheat root volume and water distribution in the media.

The main concern is that, from my point of view, NTC analyses of water distribution and images from roots submitted to Cd+drought are missing.

Comments on the Quality of English Language

- In the abstract, lines 15-16, “drought stresses” must be changed to “drought stress”…

 - Define ICP-MS, line 149…

- Line 196 must be modified “…chlorophyll content, measured as the special products analysis division (SPAD)…

Reviewer 2 Report

Comments and Suggestions for Authors

The authors deal with the response of two durum wheat genotypes to the combination of Cadmium and drought stress on shoot and root. Experiment is conducted under controlled conditions. The novelty is represented by the use of non-invasive neutron computed tomography (NCT). Two main critical methodological aspects emerge from the manuscript: the reduced experimental duration (22 days) and the absence of the drought control without Cadmium, that makes incomplete the interpretation of the results.

I am not happy to consider these limitations critical. Further, limited physiological measurements were carried out to make the observations consistent, beyond root development.  It is clear that this is a preliminary study for further experiments. Good luck.

Reviewer 3 Report

Comments and Suggestions for Authors

Review-plants-2848321

Comments to authors

The authors' objective to investigate the effects of cadmium and drought stress on the development of wheat varieties is good, but the methodology, results and conclusion chapters need improvement. Some of the figures need to be rearranged, significant differences between treatments need to be checked and explanations need to be completed. 

Method section

What was the soil clay loam or sandy-loam? What were the characteristics of the sandy soils? -Instead of the Swedish soil the soil type and properties should be given here.

-Line 426: Transpiration rate (E) was measured, but this data is not presented. In the discussion chapter, however, it is mentioned in lines 254 and 258-259 respectively. It should be corrected. 

Results

-Fig.1 (a,b,d) shows the effect of Cd dose of 60 and 120 mg/kg on the A/Ci, stomatal conductance and WUE. Does this sows the data for both wheat varieties? If yes then Fig1c is not relevant as it only shows the change in morphology of the Tramadur variety to a high Cd dose (120mg/kg). I suggest that this be presented together with the other Duramonte variety in a figure so that the differences between the varieties can be judged.

-Figure 4, which shows the root volume of the varieties on different soil types, should be presented close to the aforementioned figure to compare the effect of Cd stress on soil types. At what time was the root analysis of the varieties performed at 14 DAS or 16 DAS? To be reported in the comment below the figure.

-Line 141: typing error: 1000 µmol m-2 s-1 is correct.

-Why are only the reactions of the Tramadur variety shown in Fig. 2-5? This should be justified.

-In Figure 2b, the notation of significant differences is not clear. Question is, how does the effect of Cd and CdD stress on root and shoot dry matter content compare to the control on separate soil types or to the effect of all treatments on both soils?

-In Fig. 2b, the dry matter content in the shoot under CdDSoil was as low as under CdDSand. I believe that on the same soil type, at the shoot, under CdDSoil, the difference should be denoted by the letter c and not bc when compared to the CdSoil treatment. Moreover, in sandy soils, the dry matter content of the shoot under CdSand would be better indicated by the letters ab only, as it seems to be higher than that of CdDSand. Check and correct all markings on the diagrams during all treatments.

-Line 157: „No significant differences as related to the treatment were observed for root biomass” Are you sure? Root dry matter content appears to be significantly lower under CdDSand compared to the control (CSand) (see fig2b).

-Under Fig. 3 in the note is the Cd concentration 60 or 120 mg/kg? To be published.

-Lines 186-187: This sentence is not clear. What do you mean? This is not what you can see in Figure 4a. Check and correct it.

-Line 212 under Fig. 5: What was the cadmium concentration 60 or 120 mg/kg? To be provided.

-Fig. 5 shows the response of Tramadur to Cd stress. How was the Duramonte variety? This should also be shown. 

Discussion chapter

-254 and 258-259 discuss transpiration, but no such data are found in the manuscript. Replacements are needed or sentences deleted or reworded.

-Lines 294-295: Is this your own result? Is the note in brackets in the right place?

-Lines 311-312: This seems to be an assumption and is not confirmed in the study.

-Line 325: "Lilla Harrie Valskvarn communication" should be deleted. Personal communication should not be included in the publication (see Plants journal website). 

-Conclusion should be limited to the results presented here e.g. in rows 503-504 the statement is not based on the results presented.

-Address of the number 29 in the reference list is incorrect, it should be corrected.

Comments on the Quality of English Language

Moderate editing of English language required.

Round 2

Reviewer 1 Report

Comments and Suggestions for Authors

The authors have made the suggested corrections. They explain why additional NCT analyses of Cd-drought-treated plants were not done, which would have made the study more relevant. If the editor has no problem with those analyses not being included, I suggest that the article be published in its current form.

Reviewer 3 Report

Comments and Suggestions for Authors

review plants-284321 revised

The authors have corrected the manuscript, and I accept the explanations of the controversial parts and questions in the manuscript. In its current form it will be suitable for publication.